# Teaching Life Skills in Physical Education within Different Teaching Traditions: A Narrative Review

**Benoît Lenzen \*** **, Yoann Buyck and Axelle Bouvier**

Faculty of Psychology and Educational Sciences, University of Geneva, 1205 Geneva, Switzerland;
yoann.buyck@unige.ch (Y.B.); axelle.bouvier@unige.ch (A.B.)
**\*** Correspondence: benoit.lenzen@unige.ch

**Abstract:** This narrative review of the latest papers on life skills development in physical education (PE) aims to identify and discuss different types of life skills programmes in PE in relation with four PE teaching traditions (PETTs), their variation across contexts (e.g., primary school, secondary school, teacher education), and the inherent tensions between the teaching and learning of subject knowledge and the development of life skills. We have carried out an identification and selection process close to those used in systematic reviews, resulting in 13 papers meeting all the inclusion criteria. These peer-reviewed articles show that teaching life skills in PE covers a wide range of possible combinations between three types of programmes (isolated, juxtaposed, or integrated) and the respective PETTs in which they are rooted (sport-techniques, health education, values and citizenship and/or physical culture education). These combinations as well as the (groups of) life skills targeted in the 13 programmes we reviewed do not seem to depend on the contexts in which they are implemented. The relationship between subject knowledge, in particular sports/motor skills, and life skills is approached differently depending on the combinations that have been identified.

**Keywords:** narrative review; physical education; life skills; teaching traditions



## 1. Introduction

The interest in life skills development through sport and physical education (PE) has been perceptible for the last twenty years [1–3]. Life skills have been defined by the World Health Organisation as "abilities for adaptive and positive behaviour, that enable individuals to deal effectively with the demands and challenges of everyday life" [4] (p. 1) and paired to reveal five main life skills "areas": decision making—problem solving; creative thinking—critical thinking; communication—interpersonal relationships; self-awareness—empathy; coping with emotions—coping with stress. However, other international organisations such as the UNICEF [5] as well as several academics [2,6] agree on the difficulty of defining them:

> While the concept embraces a wide range of skills and has a virtue of linking personal and social skills to the realities of everyday life, it suffers because it is difficult, and potentially contentious, to determine which skills are relevant for life and which are not. This is problematic because if all skills are indeed relevant for life, then the concept has little utility [5] (p. 8).

Moreover, when referring to personal and social development, different terms are used interchangeably to describe similar concepts [6,7], e.g., life skills, psychosocial skills, transferable skills, soft skills, socio-emotional skills, and twenty-first century skills. Despite these debates on what life skills are and how they are labelled, educational and governmental organisations have highlighted that these skills are important for adolescents' health, well-being, and their educational and occupational success [3], encouraging policy makers to integrate life skills education into the school curriculum. Results of a systematic review analysing evaluated school-based life skills programmes regarding age-specific targeted

life skills showed that programmes were mostly implemented in adolescence, and that the targeted life skills shifted from a more behavioural-affective focus in childhood to a broader set of life skills targeted in adolescence [7].

Sports and PE are seen as settings that can enhance participants' life skills [3,8]. The potential of these settings to teach life skills is grounded on several reasons [6,8,9]: sports and PE are social in nature; there is an apparent similarity between the mental skills needed for successful performance in sports and in non-sports domains; many of the skills learned in sports and PE settings can be transferred to other life domains; sports make up a setting that emphasises training and performance, similar to school and work; sport skills and life skills are learned in the same way, i.e., through demonstration, modelling and practice; sports is a significant factor in the development of adolescents' self-esteem and perception of competence; there is an apparent conceptual similarity between the philosophy of Olympism and the notion of teaching transferable skills through sports. Different teaching approaches (e.g., sports education, cooperative learning) have been shown to help PE students develop their teamwork, communication, problem solving and decision making, leadership, and social skills [3]. A review of 88 studies using several study designs, methods, and instruments to investigate a variety of concepts related to personal and social development within the context of PE and sports led to the identification of 11 themes by grouping similar concepts [6]: work ethic; control and management; goal-setting; decision-making; problem-solving; responsibility; leadership; cooperation; meeting people and making friends; communication; and prosocial behaviour. Transferability is central to the definition of life skills (WHO, UNICEF). However, the transfer of life skills from the sports or PE environment to other areas of life has yet to be operationally defined and addressed [8,10–13]. Evidence from the recent literature in sport pedagogy or psychology shows that:

- The individual learner is the critical agent in the transfer process, which occurs when they interact with potential transfer environments [11];
- Teachers and coaches should be explicit in drawing connections as to how life skills can transfer from sports to other settings [3,10–12,14,15].

Three types of life skills programmes developed for sports and PE have been distinguished [8]: (a) programmes that teach life skills in classroom settings using sport metaphors (which we refer to as isolated); (b) programmes teaching life skills in youth sport settings in addition to sport skills (which we refer to as juxtaposed); and (c) programmes teaching life skills within the practice of PE and sports at the same time with physical skills (which we refer to as integrated). Among the programmes described by Goudas to illustrate his categorisation, the GOAL (Going for the Goal) programme [16] fell within the first category. This programme was designed to teach adolescents a sense of personal control and confidence about their future. It consisted of 10 sessions taught by selected and trained high school students to middle school or junior high school students. The SUPER (Sports United to Promote Education and Recreation) [17] programme was classified in the second category. This programme was taught in a manner similar to sports clinics, with participants involved in three sets of activities: learning the physical skills related to a specific sport; learning life skills related to sports in general; and playing the sport. The third category involves modifications of existing programmes so that these are embedded within the sport or PE practice, for instance, an abbreviated version of the SUPER programme [9]. This team-sports-based programme comprised three life skills: goal setting, problem-solving strategies, and positive thinking. The integrated nature of the programme is illustrated by the following situation. In several sessions, students were taught a three-step procedure for problem solving; then, they were presented with modified basketball and volleyball games requiring a novel solution and were asked to use the three-step procedure to find a solution.

From a didactic perspective oriented towards the study of the intertwined teaching and learning processes with a special focus on the knowledge taught [18,19], considering political and economic demands in learning and teaching PE through these contrasted

types of programmes is likely to lead to tensions between the transmission of a core of subject knowledge and the requirement to address these societal issues, particularly in terms of motor skills learning. Depending on the ways in which these new social demands are met, i.e., isolated, juxtaposed, or integrated ways of teaching life skills as well as the didactic processing [20] of social practices taken as reference [21], we assume that PE will be rooted in different teaching traditions, with this concept initially highlighting what counts as content, goals, and values for science education [22,23]. Relying on an overview of the sport pedagogy literature to identify the traditions underlying the official discourses in PE, four broad educational directions for PE have been distinguished [24]. In the PE teaching tradition (PETT) "Teaching PE as sport-techniques", PE content typically includes sports-specific movements or more generic and fundamental skills such as throwing, catching or kicking a ball. A hierarchical order from simple to complex elements is favoured and students have to master the easier skills before being confronted with the most advanced. Emphasis is generally placed on the surface features of motor techniques. The PETT "Teaching PE as health education" is based on the idea that PE should teach students to manage their physical activity and develop healthy lifestyles. PE is seen as a possible solution to the increasingly sedentary lifestyle, obesity, cardiovascular problems, etc., even if the link between PE and lifelong physical activity still needs to be firmly established, and the normativity of such an educational project faces many criticisms. According to the PETT "Teaching PE for values and citizenship", the main objectives of PE are to teach students values such as self-responsibility, respect for differences, conflict resolution, and participation in the democratic class environment. Pedagogical models such as "Sport for Peace" [25] and "Teaching Personal and Social Responsibility" [26] in the USA, as well as "La République des sports (1964–1973)" [27] in France are emblematic of this tradition, which views PE as a place where political volition and the creation of today's citizens are at the heart of the teaching. Finally, the PETT "Teaching PE as physical culture education", which is still in construction and may be seen as an attempt to integrate the three previous perspectives, is not only about learning facts, methods, or how to think as a sportsperson, but it is also about being socialised into a specific view of embodied culture, i.e., "a broader corporeal discourse that is concerned with all aspects of meaning-making centred on the body" [28] (p. 98). "Teaching Games for Understanding" [29] in the UK, "Sport Education" [30] in the USA, and "Sport de l'enfant (1965–1975)" [31] in France are the pioneering pedagogical models of this tradition, all of which are rooted in such an integrative vision of physical culture.

Therefore, this literature review aims to answer the following two research questions: (a) in what types of programmes and teaching traditions are life skills taught in PE? and (b) does it vary according to the contexts (preschool, primary school, secondary school, higher education, continuing education, teacher education)? On this basis, we can then discuss the tensions between the transmission of a core of subject knowledge and the requirement to develop life skills through PE.

## 2. Materials and Methods

This study takes the form of a narrative review, i.e., a publication that describes and discusses the state of the science of a specific topic from a theoretical and contextual point of view (in this case, the didactic perspective). In general, this type of review does not describe the methodological approach that would permit the reproduction of data nor answer to specific quantitative research questions. However, "the quality of a narrative review may be improved by borrowing from the systematic review methodologies that are aimed at reducing bias in the selection of articles for review and employing an effective bibliographic research strategy" [32] (p. 230). Therefore, we have undertaken a structured methodological approach as follows. The literature search was carried out in English and French from the database Google Scholar. The English keywords used for this research were life skills, physical education, didactics, teaching, and learning. The French keywords were compétences transversales, éducation physique, and didactique (in

French, this scientific approach systematically encompasses the notions of teaching and learning). Further inclusion criteria were applied to narrow the results and make them as specific and up-to-date as possible: year of publication of the scientific contribution from 2021 onwards; peer review; full text available. A first quick reading of the 291 articles resulting from this literature search (236 in English and 55 in French) by the first author of this paper led to retain only those that were specifically concerned with the general topic of life skills development in PE. A more in-depth reading of the 32 articles resulting from this first selection (26 in English and 6 in French) by all three authors of this paper led to the elimination of those that did not provide sufficient information on the life skills programmes content and did not allow linking it to one or more PETTs ($n = 19$). Thus, out of the initial 291 papers, only 13 were found to meet all the inclusion criteria. Table 1 summarises this identification and selection process.

**Table 1.** Identification and selection process.

| Steps | Criteria | English | French | Total |
|---|---|---|---|---|
| 1. Identification | • Year of publication from 2021 onwards<br>• Peer review<br>• Full text available | 236 | 55 | 291 |
| 2. First level of selection | Specifically concerned with the general topic of life skills development in PE | 26 | 6 | 32 |
| 3. Second level of selection | Provide information on the ways in which life skills are taught in PE | 13 | 0 | 13 |

Results of the research were first divided, according to Goudas (2010)'s categorisation [8], by teaching life skills (a) in classroom settings (which we refer to as isolated), (b) in addition to physical skills (which we refer to as juxtaposed), and (c) at the same time with physical skills (which we refer to as integrated). To carry out this first analysis, we considered several criteria. The presence or absence of motor activities linked to social practices taken as reference made it possible to discriminate between the first category (isolated) and the other two (juxtaposed and integrated). The terms used in the articles to describe the life skills programmes made it possible to discriminate between juxtaposed (e.g., "beside methods of teaching formal skills/knowledge"; "in addition to targeted forms of play"; "homework readings") and integrated (e.g., "including key elements of the life skills framework"; "commonalities among motor, cognitive and life-skills intervention"; "adopting a championship format") ways of teaching life skills. This first analysis was additionally based on the examples and illustrations that were provided in some of the articles (e.g., timing and types of learning activities offered to learners).

In the second stage, we linked the life skills programmes to one or more PETTs using several criteria. Those categorised as isolated automatically excluded the PETTs "sport-techniques" and "physical culture education" because of the absence of motor learning involved. The terms used in the articles also made it possible to discriminate between the PETTs "sport-techniques" (e.g., "formal skills/knowledge"; "basic techniques for individual sports"), "health education" (e.g., "regular physical activity"; "health-related topics"; "safe living"), "values and citizenship" (e.g., "contributes to improvement of self, school, society and the world"; "addresses various societal issues and nurturing pro-social behaviours"; "TPSR-focused professional development programmes") and "physical culture education" (e.g., "these games offer an ideal relational scenario to educate on interpersonal relationships"; "to learn activities that will result in students becoming more skilled and understanding the history, traditions and nuances of the sport and becoming willing participants in sports culture"; and "to educate people about the risk inherent in circus culture and the creative process"). This second analysis was additionally based on the examples and illustrations that were provided in some of the articles, through characterisation of the didactic processing of social practices taken as reference (How

similar is what is taught in PE to the sports or games played outside school? How similar are the life skills taught to the culture of the sports or games taken as reference?).

In a third stage, we related this double categorisation to the contexts studied in the papers.

## 3. Results

A summary of the review of the 13 articles resulting from the identification and selection process is provided in Table A1 in Appendix A.

### 3.1. Types of Programmes and Teaching Traditions

The life skills programmes we reviewed are shared between isolated ($n = 3$), juxtaposed ($n = 3$), and integrated ($n = 7$) ways of teaching life skills in PE. The isolated programmes refer to the PETTs "health education" and/or "values and citizenship". The juxtaposed ones refer to the PETTs "sport-techniques", "health education" and/or "values and citizenship". Finally, the integrated ones refer to the PETTs "values and citizenship" or "physical culture education". Table 2 illustrates this distribution between eight categories and subcategories of life skills programmes in PE.

**Table 2.** Types of programmes and teaching traditions.

| Types of Programmes | Teaching Traditions | Total |
|---|---|---|
| Isolated | Health education | 1 |
| | Health education/values and citizenship | 1 |
| | Values and citizenship | 1 |
| Juxtaposed | Values and citizenship | 1 |
| | Sport-techniques/health education | 1 |
| | Sport-techniques/health education/values and citizenship | 1 |
| Integrated | Values and citizenship | 2 |
| | Physical culture education | 5 |
| Total | | 13 |

The following are examples related to each of these categories and subcategories.

We categorised as isolated and rooted in the PETT "health education" a description of the new modelled conditions of training future Ukrainian teachers to form the competence of the safe living of children [33]. Educational activities include, among others, a discussion of important issues of the protection of life from dangers of various origins (e.g., the rules of safe stay at home, on the street, on the water, on ice, on the playground, sports grounds), the exhibition of means of protection, a demonstration of popular science and documentary films.

A proposal for developing emotional competencies as a teaching innovation for higher education students of PE in Spain [34] exemplifies an isolated way of teaching various life skills, rooted in both PETTs "health education" and "values and citizenship". The programme consists of 16 activities in relation to one or more of the following contents: (1) knowledge, identification, understanding and management of emotion; (2) emotional language; (3) mindfulness of the senses and our surroundings; (4) intelligent optimism and positive emotions; (5) a critical analysis of negative emotions; (6) resolution of intra- and interpersonal conflicts; and (7) the development of social skills. The following description of a specific activity carried out during the sessions entitled "Draw your silhouette (Contents: 1 and 2)" illustrates how life skills may be taught to PE students without any reference to physical and sports activities:

> Each person with a piece of paper and a pen draws a silhouette that represents him/her on a piece of paper. Then, using coloured pencils, they paint in which

areas of the body they notice the different emotions and in which colour [34] (p. 14 of 20).

A modular training model designed in particular for coaches and PE teachers, within the framework of the "No Violence in Sport" (NOVIS) project in Italy [35], illustrates an isolated way of teaching life skills, rooted in the PETT "values and citizenship". This modular training is supplied in three macro-training units (TU). The first TU covers general topics such as relations between youth, sport clubs, and families, violence in sports, and values through sports. The second TU provides participants with didactic recommendations to create a mastery (task-involving) motivational climate in youth sports and PE and to promote inclusive education. The third TU is devoted to the implementation of multimedia didactic tools (e.g., sports charts, logbooks, videos).

We categorised as juxtaposed and rooted in the PETT "values and citizenship" a brief description of sports-for-development type of programmes delivered as PE content by non-governmental organisations (NGO) in some lower quintile schools in South Africa [36]. In this context, the NGO coaches teach the sports-to-life approach and the students apply the values learnt in PE classes to real-life situations.

In a paper focused on connections between social relationships and basic motor competences in early childhood in Switzerland [37], we found proposals aiming at preventing the exclusion of children with poor motor competence and, at the same time, creating situations in which those children experience the joy of movement and take the opportunity to improve their motor competence without feeling ashamed. These proposals, which we categorised as juxtaposed and rooted in both PETTs "sport-techniques" and "health education", suggest that:

> In this context, extracurricular measures should also be examined and developed, such as the design of schoolyards that promote physical activity or the organisation of extracurricular sports-oriented activities that can, among other things, provide a meaningful rhythm to everyday school life [37] (p. 8 of 10).

An intervention programme designed for students of elementary school (average age 14.6 years old) in Bosnia and Herzegovina [38] illustrates a juxtaposed way of teaching life skills in PE, rooted simultaneously in the three PETTs "sport-techniques", "health education", and "values and citizenship". Besides the innovation in lesson organisation, methods of teaching formal skills/knowledge (e.g., basketball, soccer, etc.), and taking care of regular physical activity, the programme focuses on the improvement and application of the growth mindset, critical thinking, and self-cultivation through methods such as constructive feedback, conversations about topics such as success/defeat/win/loss, homework readings, and mindful meditative breathing techniques.

A methodological intervention for developing respect, inclusion, and equality in PE [39] is a good example of an integrated way of teaching and learning these life skills, rooted in the PETT "values and citizenship". The intervention is designed for 20 sessions in secondary school, twice per week, with a length of 50 min each. It is divided into six different sports modalities (athletics, volleyball, basketball, football, handball, intercross) as well as a final section of popular and traditional games. The following description of a final activity in athletics entitled "Blindfolded circuit (12 min)" is indicative of a teaching of PE in which life skills are dealt with by applying a didactic processing to social practices taken as reference that enhances their educational potential:

> There are two circuits formed with diverse materials, and the students will have to complete them blindfolded (one circuit per team). It consists of a zigzag in cones, hurdle crossing (passing underneath), jumping a step with two feet, searching for a cone to place in the hoop and the final sprint along the court. In pairs, the blindfolded one is guided by the partner, and they can touch each other [39] (p. 17 of 21).

On the other hand, an integrative methodology for circus training based on the creativity and education of physical expression [40] illustrates an integrated way of teaching

and learning creativity and risk-taking inherent in circus culture, rooted in the PETT "physical culture education". For teachers and educators as well as the general population, this training programme consists of cycles of one trimester (12 weeks). The following description of a 20 min skill learning session aimed at mastering basic exercise for safety and autonomy illustrates how life skills are taught at the same time with skills representative of circus culture:

> This section begins with a sequence of preparatory or pre-acrobatic movements, followed by at least three variations of practice: repetitions of the most effective technical patterns; a directive/guided exercise for exploring different expressive dynamics; and a game of creating a dramatic composition for the movement actions [40] (p. 510).

### 3.2. Contextual Variations

An overview of the connections between contexts (preschool and/or primary school, secondary school, and other contexts), types of programmes, and teaching traditions is presented in Figure 1.

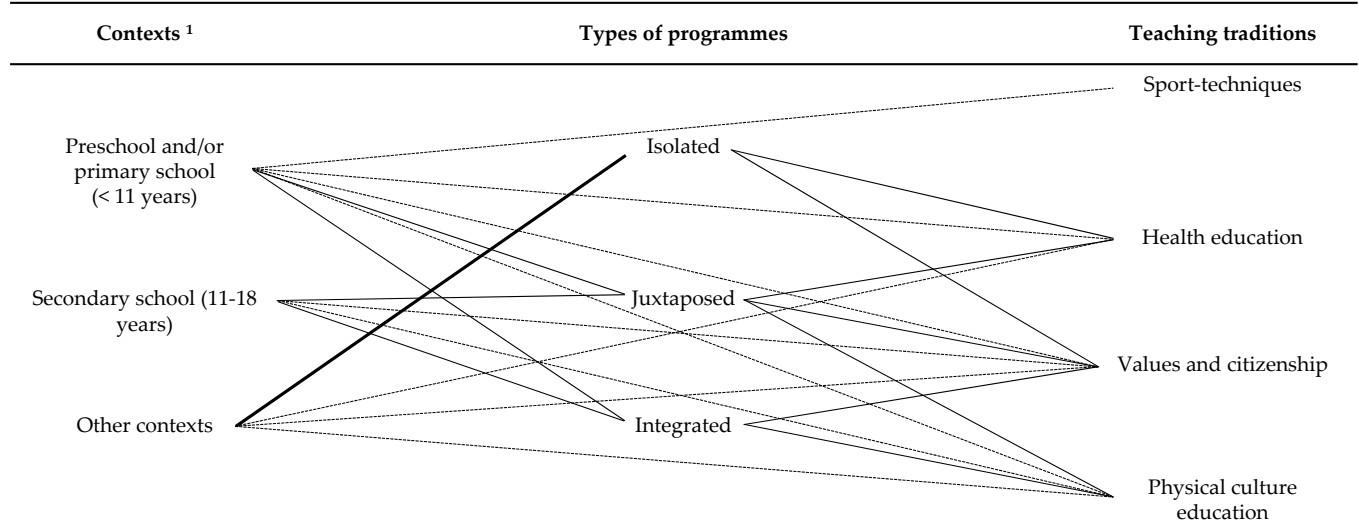

¹ As terminology and ages vary across countries, we set the age limit between primary and secondary school at 11.

**Figure 1.** Overview of the connections between contexts, types of programmes, and teaching traditions.

This overview shows little variations across contexts, except that: (a) all three programmes categorised as isolated come from studies conducted in a higher or teacher education context (as indicated by the bold connector) and (b) life skills programmes implemented in secondary schools are only rooted in the PETTs "values and citizenship" or "physical culture education", whereas those implemented in primary schools are rooted in all four PETTs. Regarding the first difference, it might seem normal that future PE teachers are not directly confronted with sports practices during training units dedicated to life skills education. However, the concept of pedagogical isomorphism and other findings of this review suggest that this is not necessarily a suitable training strategy:

> Finally, it is important to acknowledge PDFs (professional development facilitators) play a pivotal role in delivering professional development programmes and supporting teachers learning. For example, in some cases, professional development programmes are delivered in a way where PE teachers do not actively engage with course materials, which can create a disconnect between theory and practice [15] (p. 13).

The second difference could be put into perspective with the results of studies showing that the targeted life skills vary depending on the age of the students [7]. However, the

numerous and varied (groups of) life skills addressed in the 13 programmes we reviewed (when specified) do not seem to vary between the contexts. They broadly correspond to the WHO's five main life skills "areas" and to the majority of the concepts related to personal and social development within the context of PE [6]. Table 3 lists the life skills identified in the articles dealing, respectively, with our three categories of context.

**Table 3.** Life skills taught in different contexts.

| Contexts | Life Skills |
|---|---|
| Preschool and/or primary school (<11 years) | cooperation (prosocial), empathy (prosocial), quick-temperedness (antisocial), disruptiveness (antisocial), collaboration, problem solving, conflict resolution, working together. |
| Secondary school (11–18 years) | growth mindset, critical thinking, self-cultivation, various societal issues, prosocial behaviours, social inclusion, respect, equality, relational well-being, emotional well-being. |
| Other contexts | knowledge identification, understanding and management of emotions, emotional language, mindfulness of the senses and surroundings, intelligent optimism and positive emotions, critical analysis of negative emotions, resolution of intrapersonal and interpersonal conflicts, development of social skills, personal and social responsibility, conscious attitude towards own lives and health, mastering the basics of a healthy lifestyle, life skills of safe and healthy behaviour, creativity, risk, violence in sport (antisocial). |

Finally, it may be noted that all five programmes rooted in the PETT "physical culture education" rely on collective activities (team games, relay running, circus) (see Table A1 in Appendix A). The latter are processed in a didactic way, emphasising life skills inherent in their culture [41]: social inclusion, relational and emotional well-being, cooperation and empathy (prosocial) vs. quick-temperedness and disruptiveness (antisocial) in team games; collaboration, problem solving, conflict resolution, and working together in relay running; risk (of daring to act in a dialogue with spectators) and creativity (resulting in circus performances for an audience) in a circus.

## 4. Discussion

In line with the principles underpinning this type of study [32], this narrative review aimed at describing and discussing the state of the art of (a) the long-standing issue of life-skills development through sports and PE, together with (b) the contemporary framework of teaching traditions in PE. We consider that this combination is relevant from a didactic point of view, as it allows us to inform about the tensions between the transmission of a core of subject knowledge and the requirement to develop life skills through PE, which is little documented in the literature. In a field close to that of life skills development, tensions of this nature are highlighted in relation to the use of pedagogical models such as sports education and cooperative learning. In these contexts, student pedagogical interactions are often characterised by positive cooperation and engagement and are generally efficacious in developing participants' knowledge of lower complexity content learning goals. However, they often fail to facilitate the learning of a higher order content, primarily due to deficiencies in student coaches' ability to elaborate content through appropriate demonstration, error diagnosis, and task modification [42–44].

According to the PETTs framework, what is considered as subject knowledge may considerably vary according to the different PETTs [24], and the boundary between subject knowledge and life skills can appear to be shifting. Considering that the time allocated for PE in the school timetable is often limited [8,45,46], teaching life skills in isolation inevitably takes away from the time that could be devoted to the teaching of motor skills, which

remain at the core of learning in PE teaching grounded in the PETTs "sport-techniques" and "physical culture education". In this review, isolated ways of teaching life skills in higher and teacher education contexts are thus rooted in the PETTs "health education" and/or "values and citizenship". By analogy, teaching life skills in this way in a school context would reinforce the learning specific to these orientations, i.e., learning to use their knowledge to solve different kinds of problems, such as taking care of themselves and their health, and/or developing engagement and willingness to interact positively with others [24] at the expense of the acquisition of sports/motor skills [8].

In juxtaposed life skills programmes in PE, the skills taught are supposed to be closely related to sport, which is beneficial for both performance enhancement and for well-being in life, while special effort is applied to stress the transferability of skills [8]. In this review, this type of programme has been associated with the PETTs "sport-techniques", "health education", and/or "values and citizenship". The relationship between performance/motor skills and life skills was variously addressed in the three corresponding articles. In a research studying the connections between social relationship and basic motor competencies in early childhood [37], results clearly showed such a connection, but due to the cross-sectional study design, it was not possible to identify the direction of causality. However, according to the authors:

> The current state of research provides indication that early experiences playing have an impact on children's later ability to integrate themselves into a community and that persons with high motor competences are more likely to be able to participate in the culture of sports and movement during their lifespan and are, consequently, also more physically active than persons with low motor competence [37] (p. 8 of 10).

In a national study on the state and status of PE in South African public schools, different pedagogical approaches emerged and were identified as: (a) a sports-focused approach, (b) a health-focused approach, (c) a life-skill focused approach, and (d) a non-teaching or self-learning approach [36]. We have described the third approach, which is of particular relevance to our study, in the results section, highlighting its roots in the PETT "values and citizenship". Due to the research design of this study, we cannot establish a relationship between this particular approach and the multiple benefits of PE reported in this study as perceived by the Heads of Department, teachers and students, falling within themes associated with health, physical or motor-related aspects, social, and psychosocial or psychological benefits. Similarly, we cannot associate that life-skill focused approach with the author's observation that teaching sports skills out of context of practicing or playing the sport or adapted games limited authentic learning and the transferability of personal and collective outcomes. Finally, the intervention programme in Bosnia and Herzegovina [38] that we also described in the results section showed positive effects on intrinsic motivation, self-determination, the achievement of goals, flow, thriving, and mindfulness. Motor progression was not considered in this study.

The rationale for incorporating life skills teaching within sports practice (which we refer to as integrated) is grounded on three reasons [8]: (a) life skills teaching should be easily implemented in order to be attractive to physical educators and coaches; (b) life skills teaching should require minimum time for implementation; and (c) life skills teaching should not be taught at the expense of sport skills. In this review, integrated ways of teaching life skills in primary school, secondary school, teacher education, and continuing education contexts are rooted in the PETTs "values and citizenship" or "physical culture education". We associated with the first tradition two life skills programmes based on Donald Hellison's personal and social responsibility model (TPSR), whose main objective is to achieve a teaching methodology that can convey values and skills in the lives of youths at the risk of exclusion [26]. In the first paper [39], the main objective was to design a methodological proposal for compulsory secondary education, easily adaptable to different sports and educational levels and development. The structure and functioning of the sessions derived from the structure of the TPSR session: (1) relational time; (2) awareness

talk; (3) physical activity plan; (4) group meeting; and (5) reflections. As illustrated in the results section by the "Blindfolded circuit (12 min)", physical activities offered to students could be far different from the social practices taken as reference [21], i.e., physical activities and sports that students are likely to practice or observe outside of school. This didactic processing enhancing the educational potential of the latter as well as the short periods dedicated to each of them (eight different sports and games during 20 sessions) are indicative of the priority given to values and citizenship over sports/motor skills. Indeed, the continuity of teaching and learning in classroom actions [47] lay much more in the values of respect, equity, and inclusion worked throughout the sequence than in motor skills and the cultural dimensions of the numerous sports and games taught. The second paper aimed at providing strategies for professional development facilitators (PDFs) working with PE teachers within TPSR-focused professional development programmes [15]. Little guidance was supplied on how to teach motor and social skills together, but the authors stated that "implementing TPSR is not incompatible with motor objectives, assessment moments, and performance outcomes in school sport" (p. 16). The question of transferring acquired behaviours from the activity environment to other areas of life was given importance. The proposed strategies included a self-assessment tool for PDFs so that they could understand which TPSR contents were developed within the professional development programme, and this tool featured an item relating to transference.

The other five integrated programmes were thus rooted in the PETT "physical culture education". We have already highlighted in the results section that they have the particularity of being based on collective activities (team games, relay running, circus) (see Table A1 in Appendix A). In line with the characteristics of this tradition, more explicit links are made in these articles between the motor, cultural, and educational dimensions of the activities taught, underlying the value of collective activities in this respect:

> A context that may help capitalize on commonalities among theory-based interventions in PE to create an integrative approach is that of team sport games, which seems best suited to combine motor skill learning, cognitive stimulation, and life-skills education [14] (p. 2 of 23).

> These games offer an ideal relational scenario to educate on interpersonal relationships [48] (p. 8 of 19).

During a seven-session pedagogical intervention based on a championship using the Marro (Prisoner's Bar) game, the characteristics of the Marro League obliged the participants to consider two aspects: (a) the objective of the game, scoring according to the outcome of the game; and (b) relationship with peers or opponents, a subjective score on the level of competence of the students aimed at educating aspects such as self-esteem, empathy, respect, effective communication, and others [48]. In another paper aiming to understand students' experiences and behaviour towards social inclusion—such as passing the ball—in team activities and how the teacher facilitated the learning of social inclusion [49], researchers argued for a more integrative approach than the one they had observed. Indeed, learning to pass the ball through external control (i.e., teacher's instruction) is not the same as understanding why one should pass the ball. Therefore, PE teachers should consider the behavioural (passing the ball), cognitive and social (understanding why one should pass the ball), and emotional aspects (desire to pass the ball) of learning. Thus, such an integrative vision of teaching and learning life skills in PE, shared by these last five articles, embraces the principles of the PETT "physical culture education". It is likely to result in students "becoming more skilled and understanding the history, traditions and nuances of the sport and becoming willing participants in sports culture" [50] (p. 2600 of 2606). This expectation is confirmed by the results of the three quantitative studies in this subcategory, showing the positive effects of integrative pedagogical strategies in terms of students' cognitive learning outcomes on the concept of relay running [50], motor competence, cooperation and empathy [14], the reduction of motor conflicts, and the intensity of negative emotions [48]. On the other hand, the main result of the qualitative

experimentation of an innovative methodology for circus training was to discover that the introduction to this artistic activity can be approached creatively, even if the learner has no prior repertoire in circus training. The participants' experience reports revealed a broad understanding of the characteristics of this artistic language. In practice, they showed the learning of body expression skills, circus techniques, and notions of circus creative practice progressions. This approach proved effective in overcoming beginners' resistance to creative practices, who embraced the challenge and risk of daring to act and having a 'skin deep' interaction with the spectator [40].

From these last programmes, we may wonder whether life skills (if considered as such) so embedded in the culture of the activities taught can be transferred to other areas of life and if so, how. Of the 13 papers we finally selected for this review, only four address the issue of the transferability of life skills [14,15,36,39]. In one of them showing no transfer, the life-skill-focused approach was not involved, but rather, the health-focused approach [36]. In the other three studies, strategies to strengthen the transfer (e.g., explicit discussion and deliberate practice of life skills) were included in the intervention design, but the transferability of life skills was not among the dimensions assessed [14,15,39]. Of course, this is rather unfortunate given that life skills programmes are based on the assumption that the skills learned can be transferred to other settings in life [8]. Apart from the difficulty of conducting longitudinal evaluations that track youth over time and measures that examine if life skills learned in sports are indeed transferring to non-sports settings [2,8], this is probably due to the methodology of our review, which led us to highlight its strengths and limitations. As our interest was focused on the types of life skills programmes in relation with PETTs, and on the tensions between the teaching and learning of subject knowledge and the development of life skills, we chose the key words (related to the field of didactics) and defined the inclusion criteria (e.g., very recent research) accordingly. This methodology allowed us to obtain representative studies of different ways of teaching life skills in PE and to discuss the learning opportunities that these contrasting ways are offered to learners. Perhaps less restrictive inclusion criteria would have resulted in studies documenting (a) isolated programmes in school contexts, whereas the studies we did select covered only other settings, and (b) programmes implemented in secondary schools rooted in the PETTs "sport-techniques" and "health education", whereas those we reviewed were only rooted in the other two teaching traditions. Indeed, the literature shows that "Teaching PE as sport-techniques" is a teaching tradition that is still disseminated in Western countries [24,28], and that life skills education promotes health-related self-regulation, especially in adolescence [7].

## 5. Conclusions

With these limitations in mind, the current narrative review adds to the existing literature by cataloguing the most up-to-date approaches of life skills development implementation in PE, analysing how they vary according to context, and discussing the inherent tensions between the teaching and learning of subject knowledge and the development of life skills. The latest empirical research and teaching proposals covering contexts as varied as preschool, primary school, secondary school, higher education, continuing education, and/or teacher education show that teaching life skills in PE covers a wide range of possible combinations between the three types of programmes categorized by Goudas [8] and the four teaching traditions identified in PE by Forest et al. [24]. These combinations as well as the (groups of) life skills targeted in the 13 programmes we reviewed do not seem to depend on the contexts in which they are implemented.

The relationship between subject knowledge, in particular sports/motor skills, and life skills is approached differently depending on the combinations that have been identified. Isolated programmes exclude opportunities for motor learning while integrated programmes rooted in the PETT "values and citizenship" prioritize social, psychosocial, or psychological benefits over the learning of motor skills and cultural dimensions of the sports and games taught. Further research is needed to deepen the understanding of

this complex relationship between the acquisition of sports/motor skills and life skills development, including the life skills transfer process.

**Author Contributions:** Conceptualisation, B.L. and Y.B.; methodology, B.L. and Y.B.; validation, Y.B. and A.B.; formal analysis, B.L., Y.B. and A.B.; investigation, B.L.; data curation, B.L., Y.B. and A.B.; writing—original draft preparation, B.L.; writing—review and editing, B.L., Y.B. and A.B.; visualisation, B.L.; supervision, B.L. All authors have read and agreed to the published version of the manuscript.

**Funding:** This research received no external funding.

**Institutional Review Board Statement:** Not applicable.

**Informed Consent Statement:** Not applicable.

**Data Availability Statement:** Not applicable.

**Conflicts of Interest:** The authors declare no conflict of interest.

## Appendix A

**Table A1.** Summary of the review of the 13 articles.

| Reference | Contexts [1] | Countries | Type of Programme | Life Skills | Social Practices Taken as Reference | PETTs |
|---|---|---|---|---|---|---|
| 1. Brankovic and Badric, 2021 [38] | Elementary school (14.6 years) | Bosnia and Herzegovina | Juxtaposed | Growth mindset; Critical thinking; Self-cultivation | Basketball; Soccer | Sport-techniques; Health education; Values and citizenship |
| 2. Burnett, 2021 [36] | Primary (13.5 years) and secondary (17.6 years) school | South Africa | Juxtaposed | Various societal issues/prosocial behaviours | Unspecified | Values and citizenship |
| 3. Condello et al., 2021 [14] | Primary school (10–11 years) | Italy | Integrated | Cooperation (prosocial); Empathy (prosocial); Quick-temperedness (antisocial); Disruptiveness (antisocial) | Team games | Physical culture education |
| 4. Fenanlampir, 2021 [50] | Elementary school (10–11 years) | Indonesia | Integrated | Collaboration; Problem solving; Conflict resolution; Working together | Relay running | Physical culture education |
| 5. Fernández-Gavira et al., 2022 [34] | Higher education | Spain | Isolated | Knowledge identification; Understanding and management of emotions; Emotional language; Mindfulness of the senses and our surrounding; Intelligent optimism and positive emotions; Critical analysis of negative emotions; Resolution of intra- and interpersonal conflicts; Development of social skills | Physical activity and sports in the natural environment | Health education values and citizenship |
| 6. Herrmann et al., 2021 [37] | Preschool (4–7 years) | Switzerland | Juxtaposed | Cooperation; Problem solving | Unspecified | Sport-techniques; Health education |
| 7. Hovdal et al., 2021 [49] | Secondary school (13–15 years) | Norway | Integrated | Social inclusion | Team games | Physical culture education |
| 8. Muñoz-Llerena et al., 2022 [39] | Secondary school (11–16 years) | Spain | Integrated | Respect; Equality; Inclusion | Athletics; Volleyball; Basketball; Football; Handball; Intercross; Popular/traditional games | Values and citizenship |

**Table A1.** *Cont.*

| Reference | Contexts [1] | Countries | Type of Programme | Life Skills | Social Practices Taken as Reference | PETTs |
|---|---|---|---|---|---|---|
| 9. Rillo-Albert et al., 2021 [48] | Secondary school (14–16 years) | Spain | Integrated | Relational well-being; Emotional well-being | Marro (Prisoner's Bar) Stealing stones; Dodgeball; Pass the Treasure | Physical culture education |
| 10. Santos et al., 2021 [15] | Teacher education | Portugal Ireland | Integrated | Personal and social responsibility | Unspecified | Values and citizenship |
| 11. Savchuk et al., 2021 [33] | Teacher education | Ukraine | Isolated | Conscious attitude to their own lives and health; Mastering the basics of a healthy lifestyle; Life skills of safe and healthy behaviour | Unspecified | Health education |
| 12. Tucunduva, 2021 [40] | Continuing education (16–39 years) | Brazil | Integrated | Creativity; Risk | Circus | Physical culture education |
| 13. Vitali and Conte, 2021 [35] | Coaches and PE teachers education | Italy | Isolated | Violence in sports (antisocial) | Unspecified | Values and citizenship |

[1] We retain the terms used in the articles, specifying the age of learners where available to avoid confusion between different denominations of the same grade.

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
