# Peer review of "Teaching Life Skills in Physical Education within Different Teaching Traditions: A Narrative Review"

_education, doi:10.3390/educsci13060605_

Round 1

Reviewer 1 Report

I enjoyed reading “A Review of the Teaching and Learning of Life Skills in Physical Education”. The topic is very important. The article highlights different types of life skills programmes and points to critical issues in the teaching and learning of life skills (see comments in the paper).

However, I rejected the paper because there are many things that needs to be developed or changed.

Introduction

The notion of life skills is very quickly defined, whereas it is at the core of the study.

What are the different definitions and the different categorizations of life skills?

Goudas (2010) highlights that there is a connection between life skills definitions, life skills frameworks and the life skills teaching programmes. The authors don’t present the existing life skills teaching programmes in the literature.

The introduction focuses on the concept of PETTs, which is not at the core of the life skills framework. As you use the Goudas framework, why not relate the life skills to the life skills teaching programs or the different curricular models and then secondly to the PETTs ?

What are the relations between life skills and PETTs ? How do you relate a program to a PETT?

How do you « infer the teaching traditions and possibilities for learning offered to students » from the analyses (c) ? That is not clear.

Materials and methods

The Methods-section needs to be developed.

The contexts studied in the papers, from elementary school to PETE, are not precised and justified.

The PETTs depend on countries (Forest, Lenzen & Ohman, 2017), but the countries in which the studies are conducted are not specified.

What are the analysis criteria? How do authors categorize the programmes (juxtaposed / isolated / integrated) ? Could you detail the indicators?

These sentence questions us: « To carry out this analysis, we considered subject knowledge from the perspective of the PETT “Teaching PE as physical culture education” ». Can the researchers use a specific PETT to carry out the analysis?

Did authors use other criteria ? Some data are indeed present in the discussion : learning opportunities, risks, transfer, characteristics (steps) of the situations.

It seems possible to deepen your analysis of life skills programmes by using other criteria.

Why and how do you group the different names of life skills ? Why don’t you use the classifications of life skills and life skills teaching programmes? How do you relate life skills programmes and PETTs ?

Is it relevant to "identify which life skills are most commonly taught whereas there is only 17 articles?

How do you select the illustrations to present the results ? What criteria are used to analyse these illustrations ?

Results

This part is brief. Certain data presented in the discussion coul be moved in the results.

The first part is about « Ways of Teaching Life Skills in Physical Education ». This title is a bit vague. You should precise that it is about the relationship between life skills and physical skills.

Douglas (2010) had already illustrate these categorization of programmes (juxtaposed / isolated / integrated). What do you bring that is new?

The results (3.2. Life Skills Taught in Physical Education) are meagre.

Discussion

The first part of the discussion (description of programmes) includes results with illustrations and citations.

You should present the literature about risks in the introduction and then discuss the results in relation to the literature review.

The researchers interpret the results from the perspective of the PETT “Teaching PE as physical culture education” (« a distortion of the physical and sports activities taught ») and make recommendations :

« A promising avenue to avoid the risk of neglecting PE specific knowledge when trying to teach life skills lies in conducting an epistemological reflection about the core principles of the social practices taken as reference in PE ».

You should describe the contents of life skills programmes by highlighting the relationships to the different PETTs without judgments (transformation instead of distortion for example, prioritize life skills over physical skills instead of neglecting PE specific knowledge).

Reviewer 2 Report

Introduction

·         There exist several reviews in this area and the authors should include those in the introduction and frame how this new review adds to the knowledge base on life skills in PE.

·         The review period is quite narrow 2021 (but still there are papers from 2020 and 2022). This selection should be clarified.

·         Gouda’s name is referred to directly in the aim, but it could be stated more clearly when presenting the three types of life skills programs.

·         There exist a great deal of research on the transfer of life skills from sport to others parts of the life, which the authors present in the end of the discussion. It would suggest to bring this up front as it relates to the three different program types.

Methods

·         There exist several types of reviews. Without being an expert, I would suggest to frame this review as a narrative review, which gives the opportunity include authors perspectives on the subject.

·          Contexts are not mentioned anywhere in the manuscript besides in Table 2. I consider it pretty important to understand the different approaches to life skills in PE. An don’t understand the contexts “literature search” and “pandemic”.

·         There is a cross reference to the TPSR model to the introduction, but to me it is not obvious what it is refereeing to.

·         In general, the method sections should be more clear and detailed in the categorization processes. E.g. how many full text reads did the authors do, did the authors independently do the review and categorizations, which criteria was related to categorization.

Results

·         No need for Figure 1

·         I would suggest adding the third aim to the result sections as well. The authors must be more clear in the argument of how the different programs relate to different teaching traditions – and how this analysis is based on the empirical data from the papers.

·         I would like a more comprehensive presentation of the life skills covered (section 3.2). In my opinion the list of life skills in Table 2 is very different and sometimes not clear if it really is a life skill. Did you have any criteria to what it should take before it could be considered a life skill.

Discussion:

·         Overall, the discussion is very fragmented and hard to follow. If the third aim was part of the result sections and the discussion rewritten it might help.

·         The first section starting with “As evidence..” should be places in the end of discussion and related to the quality of the reviewed papers.

·         Don’t understand the connection with the external providers (ll. 269-281)

·         The risks raised related to life skill programs (ll. 306-309) is relevant. The discussion could be improved. For example, there could be other issues than “motor competences” relevant for PE according the teaching traditions, which one might miss. I don’t understand the epistemological reflections and the meaning of it (ll. 320-325). The discussion of the transfer seems quite shallow. Could the review papers be more integrated in the discussion. And all taken together I consider the conclusion to be too positive to the integrated way.

·         I miss a strength and weakness section, which elaborate on the methods of the review and the included papers.

Conclusion

·         The conclusion should be elaborated and try to answer the three aims of the paper in a nuanced way integrating the results.   

Reviewer 3 Report

Thank you for the opportunity to review the paper.

I appreciate the holistic research question to investigate life skills in physical education as life skills could be considered as a prerequisite for participation in physical education (e.g., team sports). Overall, the objective of the article is not clear. In the introduction you start well and name different methods to teach life skills into PE. However, the method is unspecific, the keywords are not differentiated, only google scholar is presented as a database and it is not explained why only studies since 2021 are integrated. I would recommend to fundamentally revise and restructure the article. The reader should know what life skills are and why they are relevant. Then, studies can be presented that examine life skills in PE. Later, it should be classified in which form these can be taught and integrated. In addition, the literature research should be systematic and based on the PRISMA statement.

My comments are listed sectionally below.

Abstract

The questions in the abstract do not agree with the questions in the text (l. 79-82).

 Introduction

-     l. 29: I would recommend citing the original reference of the WHO and UNICEF

-       there are 10 life skills defined by the WHO. I would recommend listing them.

-    l. 50: it would help to number the four broad educational directions for PE with first, second, …

-    l. 77: please rephrase the question into a statement.

-       As the TPSR model is not sufficiently explained, it is somewhat confusing as it is explicitly referred to here.

Method

-       Google Scholar is not relevant as a database - scientific databases such as ERIC, SPORTDiscus, Web of Science and Scopus would have to be used here.

-       l. 113ff: The keywords are too unspecific. Concrete life skills could be named here based on the theoretical concepts.

-       In this form, many more literature should come up. It must be shown how the keywords were used together (AND, OR; WITHOUT etc.).

-       l. 87/l.96: are filters the same as inclusion criteria? If yes, I would recommend to consistently use “inclusion criteria”.

-       You may consider including a PRIMSA Flow Chart as this illustrates the phases of a literature review (http://prisma-statement.org/PRISMAStatement/FlowDiagram.aspx)

-       Table 1: I do not understand the combination of 19 life skills programs and the 17 papers that met the inclusion criteria.

-       l. 107: “based on Donald Hellison’s TPSR model (see introduction) …” à I would recommend using the abbreviation already in l. 64.

-       Please explain whether you have considered equally at studies that have conducted life skills intervention programs and studies that have investigated life skills in combination with physical education.

-       Why do you only include studies since 2021? Has a literature review on this topic been done before? The time period is definitely too short for such a contribution...

Results

-       There should be a separate chapter in which a clear list is made of which life skills were found at all.

-       Overall: To which age group do the results refer? There are certainly differences between kindergarten and secondary school in terms of life skills programs, but also in terms of the life skills that are examined in studies. I can also recommend the following article, which examines which life skills are relevant at which ages: Kirchhoff, E., & Keller, R. (2021). Age-Specific Life Skills Education in School: A Systematic Review. Frontiers in Education, 6, 221. https://doi.org/10.3389/feduc.2021.660878

-       Since the distribution is not very complex, you can dispense with the pie chart.

-       Can you clarify the presentation of the results? For example, how the life skills were assessed, or which instruments were used. Furthermore, whether there were interventions and if so, which ones?

-       113-117:This step must be made transparent, “where we felt” is not suitable as an explanation

-       l. 131 ff: You describe the studies or the program, but do not go into detail about the results. For example, what are the results of the 12-week circus training program?

-       3.1: In general, this point seems like a list of various excerpts from studies. Please reorganize the presentation of results

-       3.2: Refer here to the WHO life skills.

-       This should be placed at the front and clearly differentiated. It must also be made clear what is meant by this, and which other "terms" are subsumed here.

Discussion

Overall, the objective is not clear to me - you ask which life skills are taught, although many studies only name them at first. Accordingly, it should first be clarified which ones are understood as significant at all and then ask the question about "teaching and learning".

Reviewer 4 Report

This paper researches an interesting topic but I have some conceptual feedback. The methodology section is poor and not very rigorous.

The author(s) did not follow the content of the PRISMA 2020 declaration and its protocol, which is currently configured as one of the most used for conducting systematic reviews in the educational field, is not shown.

From my point of view,  it is necessary to review the systematic review since the beguining of the process under PRISMA protocol.

Reviewer 5 Report

Please see the attached document for reviewer comments

Round 2

Reviewer 2 Report

The paper has improved, but I have to concerns: 

- The third aim of the paper (learning experience) is not sufficiently covered in the paper (method, results, discussion and conclusion) and I will recommend to delete that aim. 

- The conclusion should be rewritten and try to answer the aims of the paper more directly and not just repeat the findings. This also applies to the abstract conclusion. 

And a suggestion: 

- I think the paper would benefit from a visual overview of the connections between the three analytical categories: context, Gouda's cat. and PETT. I image three collumns with 3-4 boxes in each with headlines and if you can draw lines between them it would create a visual overview of the analyses. 

Author Response

Dear reviewer 2,

Thank you very much for your fruitful comments and suggestions.

At the end of our introduction, we have deleted the third aim of our paper and highlighted the topic of tensions between the teaching and learning of subject knowledge, especially sport/motor skills, and the development of life skills, which was already at the core of our discussion section.

We have rewritten our conclusion section trying to answer the aims of the paper more directly. Consequently we have also rewritten the abstract conclusion.

We have followed your kind suggestion to add a visual overview of the connections between the three analytical categories (see table 3 in the second part of the results section). This overview is indeed helpful as it enriches this part of the results section dedicated to contextual variations. We have integrated these new elements in the discussion section.

Best regard.

Reviewer 4 Report

The authors have made all the suggestions and have improve the article.

Author Response

Dear Reviewer 4,

Thank you for your comments and for your positive appreciation of the major revision work we have undertaken.

Best regards.